# OpenReview forum: "Dual Mechanisms of Value Expression: Intrinsic vs. Prompted Values in Large Language Models"
_ICML.cc/2026/Conference — ICML 2026 regular_

### Official Review · Reviewer_GLu2 · 2026-03-12

**Soundness:** 3
**Presentation:** 4
**Significance:** 2
**Originality:** 2
**Overall Recommendation:** 5
**Confidence:** 3

**Summary:**

This paper studies whether value expression in LLMs arises from the same internal mechanism when values are learned intrinsically during training versus elicited by explicit prompting. Using Schwartz’s value taxonomy, the authors extract “value vectors” from residual activations and “value neurons” from MLPs, then compare intrinsic and prompted value expression mechanistically and behaviorally. The main finding is that the two mechanisms partially overlap but also contain distinct components: shared parts capture core value semantics, while intrinsic-specific components are associated with more diverse, naturalistic responses and prompted-specific components are more strongly tied to instruction-following.

**Compliance With Llm Reviewing Policy:**

Affirmed.

**Final Justification:**

The rebuttal fully addressed my concerns. The extensive robustness analyses (bootstrap validation of the decomposition, neuron threshold ablations, cross-model replication) convincingly demonstrate that the findings are not artifacts of specific methodological choices. The commitment to foreground these checks in the main text and improve figure captions will further strengthen the paper. I maintain my score of 5.

**Key Questions For Authors:**

see weaknesses

**Limitations:**

yes

**Strengths And Weaknesses:**

### Strengths
* Clear and interesting research question with strong relevance to alignment, pluralistic values, and interpretability.
* Strong empirical breadth: multiple datasets, multilingual evaluation, steering interventions, diversity analyses, and appendix experiments across additional models.
* Presentation is strong and the paper is easy to follow

### Weaknesses
* The work is rich empirically, but some methodological choices (vector extraction, orthogonal decomposition, neuron selection thresholds) may be somewhat sensitive; robustness is present but not always foregrounded enough in the main narrative.
* Figure captions are sometimes too brief and could be more informative; several figures would benefit from captions that more explicitly summarize the setup, key takeaway, and how to interpret the comparison without relying heavily on the main text.

---

> ### Author Rebuttal · Authors · 2026-03-31
>
> We sincerely thank you for your encouraging review, and for highlighting our paper's empirical breadth, relevance to alignment, and clear presentation. We appreciate your constructive feedback and will incorporate your suggestions to strengthen the manuscript.
>
> ## **Foregrounding Robustness and Methodological Sensitivity (W1)**
> We agree that robustness of our methodology should be more explicit in the main narrative. Although the paper includes a substantial set of sensitivity analyses, they are currently spread across several appendices and are therefore easy to miss. In the revised manuscript, we will foreground these checks in the main text and explicitly connect each methodological choice to its corresponding robustness analysis.
>
> Concretely, we evaluated robustness at the following stages of the pipeline:
> ### (1) **Vector extraction**
> **Token positions.** We ablated the token-pooling choice by recomputing value vectors using only the first 5, middle 5, and final 5 response tokens, and compared them against our default setting (averaging all tokens). Across all experiments, our setting consistently produced strongest steering effects (Appendix A.3).
>
> **Alternative linear estimators**. We compared our difference-in-means vectors against logistic-regression probes trained on the same residual activations. Although the probes showed high classification accuracy, their steering effects were consistently weaker than our difference-in-means method (Appendix A.3).
>
> **Layer sensitivity**. Instead of relying on a single hand-picked layer, we computed value vectors across layers and selected steering layers through a grid search over intervention strengths rather than fixing them beforehand (Appendix E.1, F.1-F.2).
> ### (2) **Orthogonal decomposition**.
> We have conducted additional bootstrap resampling analyses to rigorously validate the stability of this decomposition. Across all layer-14 value dimensions, the bootstrap-derived orthogonal vectors exhibited consistency across resamples (mean pairwise cosine similarity $\ge$ 0.989 across 1000 bootstraps) and tightly matched the original estimates (mean cosine similarity = 0.995).
> ### (3) **Neuron selection sensitivity**
> In response to your concern on neuron-selection sensitivity, we have conducted additional ablation experiments on our neuron-selection thresholds (top-15% magnitude filter, 30° angular cutoff) on Qwen2.5-7B-Instruct. By varying the magnitude filter (top-1%, 15%, 30%) and angular cutoff (15°, 30°, 45°), we confirmed our core findings are robust.
>
> First, the number of shared neurons was consistently higher than those unique to either mechanism across all settings (e.g., 5,663 at top-1% to 36,929 at top-30%), proving that **substantial overlap between intrinsic and prompted mechanisms exists regardless of threshold selection**. Also, neuron-explanation analysis across these thresholds show that **the proportion of neurons with identifiable value semantics is stable** (8.7% at top-1% vs. 5.0% at top-30%).
> ### (4) **Robustness of Evaluation**.
> We validated the automated evaluator against human annotations and repeated the comparison with alternative open- and closed-source judges. The same overall steering trends remained stable, so the conclusions do not depend on a single judge model (Appendix D.2-D.3).
> ### (5) **Robustness of the functional findings**.
> The diversity gap between intrinsic and prompted mechanisms remains under temperature sweeps, top-p sweeps, and implicit-prompt variants. This supports the view that our conclusions reflect genuine mechanistic differences rather than a narrow decoding or evaluation choice (Appendix F.5, G.2-G.5).
> ### (6) **Cross-model generalization**.
> The main behavioral and geometric findings also replicate beyond our three primary models, including Qwen2.5-32B-Instruct, Gemma2-9B-it, Qwen3-14B, and Qwen3-8B, suggesting that they are not specific to a single architecture or scale (Appendix N).
>
> These additional experiments preserve the same qualitative conclusions regarding shared/unique neuron distributions and their causal steering effects. In the revised manuscript, we will add a dedicated “**Robustness and Sensitivity**” paragraph in the main text that summarizes these checks up front, while retaining the full tables and ablations in the appendix.
>
> ## **Improving Figure Captions  (W2)**
>
> We agree that self-contained figures enhance the reading experience. In the camera-ready revision, we will thoroughly expand all figure captions. Specifically, each caption will explicitly describe the experimental setup, explain how to interpret the comparisons and axes, and clearly state the key takeaway, so readers can fully understand the visualizations without relying on the main text.
>
> Thank you again for your time, your positive evaluation, and your constructive feedback, which will greatly improve the final version of our paper.

---

> > ### Author Rebuttal · Reviewer_GLu2 · 2026-04-01
> >
> > Thank you for the detailed rebuttal. This addresses my concerns sufficiently, especially on the robustness side, and I agree that making these points more visible in the main text will improve the paper. I also think the planned revision to the figure captions will help readability.
> > Overall, I do not have further questions, and I am happy to keep my positive assessment (accept).
> > .

---

### Official Review · Reviewer_3BF7 · 2026-03-13

**Soundness:** 2
**Presentation:** 3
**Significance:** 2
**Originality:** 3
**Overall Recommendation:** 4
**Confidence:** 4

**Summary:**

This paper studies how LLMs represent and express human values. The authors compare intrinsic values, which emerge without explicit prompts, and prompted values, which are induced by value-targeting prompts. They extract value vectors and analyze neurons linked to these signals. The results show a shared component that captures core value semantics. They also find unique components tied to natural expression and instruction compliance.

**Compliance With Llm Reviewing Policy:**

Affirmed.

**Final Justification:**

The rebuttal have addressed my concerns and I'm maintaining my positive score.

**Key Questions For Authors:**

Please refer to strengths and weaknesses.

**Limitations:**

Yes

**Strengths And Weaknesses:**

### **Soundness**
**Strengths**
- The paper conducts extensive experiments on multiple models and reports consistent results across settings.
- The claims are generally supported by experiments. The analysis includes both vector-level and neuron-level evidence.

**Weaknesses**
- The neuron identification method lacks theoretical grounding. The classification relies on projections onto value vector subspaces, but it is unclear whether this reliably captures causal neuron contributions.
- The paper does not compare its neuron identification approach with alternative methods such as activation-based or gradient-based attribution. Such comparisons would help validate the robustness of the findings.
- Some behavioral evaluations remain relatively in-domain. It would be interesting to test whether steering specific values (e.g., self-enhancement) affects broader behaviors such as power-seeking ([1]) or manipulation.

### **Presentation**
**Strengths**
- The paper is clearly written and generally easy to follow. The distinction between intrinsic and prompted values is well motivated. The visualizations are helpful for understanding model behaviors.

**Weaknesses**
- The intrinsic vs. prompted setup may initially cause confusion. A clearer explanation of the prompt settings and data construction could improve readability.

### **Significance**
**Strengths**
- The researched questions over intrinsic and prompted components are interesting. Understanding how LLMs represent and express values is highly relevant to alignment and safety research.

**Weaknesses**
- The practical implications for alignment methods remain somewhat indirect. It is not yet clear how the discovered mechanisms can be reliably used to improve alignment systems. Adding insights on the guidance of human-value alignment over potential risk would help.

### **Originality**
**Strengths**
- The topic is intriguing. The authors offers an interesting perspective by directly comparing intrinsic and prompted value expression mechanisms. They reveal a potentially important link between prompted value expression and instruction compliance.

> [1] Do the Rewards Justify the Means? Measuring Trade-Offs Between Rewards and Ethical Behavior in the MACHIAVELLI Benchmark

---

> ### Author Rebuttal · Authors · 2026-03-31
>
> Thank you for your feedback.
>
> ## **Theoretical grounding and causal contributions of the neuron identification method**
>
> While our projection-based neuron identification does not guarantee causality by design, it is **theoretically motivated and empirically validated to capture causal effects**.
>
> - **Theoretical grounding**: We construct value vectors using a difference-in-means estimator, which is theoretically motivated to robustly recover linear features without overfitting to spurious correlations as supported by [1]. By averaging over 26,334 diverse input prompts, we **marginalize out prompt-specific, value-irrelevant noise to isolate the representational component consistently associated with value expression**. Consequently, subspace projection onto these causal directions provides a principled way to identify neurons aligned with value mechanisms.
>
> - **Empirical validation**: The causal contribution of these neurons is validated through intervention experiments in Section 4.3.
> We will revise Section 2.2 to explicitly mention our method's theoretical justification and empirical findings.
> ## **Comparison with alternative neuron identification methods**
> We evaluated two baselines: an **activation-based baseline** [2] (using the activation frequency gap between positive and negative samples) and a **gradient-based baseline** (using the gap in Integrated Gradients [3] scores). Steering results (https://anonymous.4open.science/r/rebuttal-0B16/results.pdf) show our projection-based method outperforms the activation-based baseline, but underperforms the gradient-based method.
>
> Nevertheless, we believe our projection-based method is better suited for our goal of investigating value expression mechanisms because, while gradient attribution identifies neurons **maximally sensitive** to promoting a target value, they are not guaranteed to be actually **involved** during value expression. In contrast, because difference-in-means is based on activated directions during value expression, this subspace is a better approximation of the components **natively utilized** during value expression.
>
> To qualitatively assess gradient-based neurons, we have conducted additional experiments applying our neuron explanation method (Appendix M) to them. Since this method derives explanations from neurons' activation patterns in natural contexts, it serves as a **direct probe into how the model natively uses these neurons**. Gradient-based neurons mostly activated in **value-irrelevant contexts (e.g., formatting or QA boilerplate)**, implying a spurious impact on value expression. This contrasts with projection-based neurons, which yielded **interpretable, value-specific explanations** (Section 5). We will add these comparisons to clarify this methodological distinction.
> ## **Broader behavioral evaluations (e.g., power-seeking, MACHIAVELLI)**
> To investigate whether steering specific values affects related behaviors, we evaluated Qwen2.5-7B-Instruct on the MACHIAVELLI benchmark [4], hypothesizing that steering toward power amplifies power-seeking, while benevolence suppresses it.
> Results confirm this: relative to the unsteered baseline, power vectors increased both power-seeking (+11.3% prompted; +6.6% intrinsic) and violation scores (+12.3% prompted; +9.0% intrinsic), which capture manipulation and intended harm. Conversely, benevolence vectors reduced these scores (intrinsic/prompted: -3.2/-3.1% power-seeking, -9.8/-7.5% violations).
>
> An additional experiment on prosociality is presented in our rebuttal to Reviewer As7k (W5).
> ## **Clarity of the intrinsic vs. prompted setup**
> To clarify intrinsic vs. prompted settings, we will **separate their pipelines clearly** in the data construction section and **add a side-by-side description of prompt designs**, including exact input and output formats under each condition.
> ## **Practical implications for alignment and safety**
> We will expand the discussion on concrete practical implications:
>
> 1. **Pluralistic Steering**: Our steering method is effective across multiple value dimensions and languages (Section 4). The increased steerability enables developers to **align models with diverse user needs and cultural contexts** without expensive retraining (e.g., parental controls in a child-facing chatbot).
>
> 2. **Monitoring Jailbreaks**: The linear representation associated with prompt compliance (Section 5.3) could serve as a useful signal for **monitoring jailbreak attempts exploiting the model's instruction-following tendencies**.
>
> [1] Nora Belrose. "Diff-in-means concept editing is worst-case optimal: Explaining a result by Sam Marks and Max Tegmark", 2023.
>
> [2] Su et al. "Understanding How Value Neurons Shape the Generation of Specified Values in LLMs", 2025.
>
> [3] Sundararajan et al. "Axiomatic Attribution for Deep Networks", 2017.
>
> [4] Pan et al. "Do the Rewards Justify the Means? Measuring Trade-Offs Between Rewards and Ethical Behavior in the MACHIAVELLI Benchmark", 2023.

---

> > ### Author Rebuttal · Reviewer_3BF7 · 2026-04-03
> >
> > Thanks for addressing my concerns. I will maintain my positive score.

---

### Official Review · Reviewer_As7k · 2026-03-14

**Soundness:** 3
**Presentation:** 2
**Significance:** 3
**Originality:** 3
**Overall Recommendation:** 4
**Confidence:** 4

**Summary:**

The paper carries an interesting premise. When constructing activation datasets for Schwartz’s ten values, which include Achievement, Benevolence, Conformity, Hedonism, Power, Security, Self-Direction, Stimulation, Tradition and Universalism, the authors extract contrastive directions in the residual stream corresponding to value expression. They then apply steering along these directions as well as ablating or amplifying influential MLP neurons associated with them. These interventions influence the behaviors of LLMs in several ways. Some effects are intuitive, such as shifting which value the model prefers when responding to situational dilemmas. Others are less expected, including changes in lexical diversity and response variability.

**Compliance With Llm Reviewing Policy:**

Affirmed.

**Final Justification:**

I quite like the technical content of the paper, and the main object of study which is the Schwarz value system (widely used in psychology), and how they express both intrinsically, and when pushed through prompting.

That said, the narrative framing is really clumsy and assumes both prior knowledge and widespread adoption of the Schwarz value system in NLP, rather than grounding these more carefully. The authors also could not adequately suggest how to make this framing more smooth, so I don't see any way to update my score to 5 or feel I have no material reservations reaining.

Ultimately, I'm still happy to give a weak accept (4), which in my opinion is fair given both the strengths and weaknesses of the paper, as well as it's reception by other reviewers. No paper is perfect, and the value outweighs the clunkiness in writing.

**Key Questions For Authors:**

1. Can you compare the Schwarz values to other previous values in the steering literature? (Both conceptually, and geometrically.)
2. What would you say is the strongest contribution of this paper?
3. How would you suggest an abstract that is more clear and concrete to digest?

**Limitations:**

Yes.

**Strengths And Weaknesses:**

### Strengths
1. I like that the paper goes beyond the oversaturated studies on refusal or sentiment directions and instead investigates value based steering vectors grounded in a psychological taxonomy. The Schwartz framework is well known in social psychology and provides a principled structure for studying value representations.
2. Figure 8, which suggests that shared value directions lie in a circular geometry corresponding to Schwartz’s theoretical circumplex, is particularly surprising and interesting. If robust, this result suggests that value representations in LLMs may exhibit meaningful structure that parallels human value theory.
3. Section 3 is written clearly and concretely, and the methodology is generally easy to follow. The pipeline of generating responses, classifying value expression, extracting difference in means activation directions, and mapping them to neurons is straightforward.
The authors also use several standard interpretability techniques to analyze these features, including logit lens style vocabulary analysis, neuron attribution, and vector steering interventions.
4. The inclusion of multiple evaluation settings such as PVQ questionnaires, situational dilemma tasks, and the Value Portrait benchmark provides some breadth of empirical evidence.

### Weaknesses
1. The abstract is overly dense and difficult to parse. It tries to simultaneously introduce the conceptual framing, mechanistic methods, and empirical findings, which results in a clunky presentation that obscures the core idea.
2. The core contribution of defining and analyzing value based steering vectors grounded in Schwartz’s value taxonomy appears relatively late in the paper. In general, the early framing feels overly focused on motivating the problem and selling the paper (which in my case was unnecessary, since the course of study is very interesting) rather than crisply stating the main contributions and technical results early.
3. More importantly, the paper lacks a clearly articulated central research question. At different points it reads like several different papers. It could be interpreted as a study of whether activation steering produces more diverse responses than prompting, a study of how intrinsic and prompted value representations differ mechanistically, or a study of the structure of Schwartz values inside LLM representations. Each of these directions is interesting but the narrative does not clearly unify them.
4. Relatedly, the causal mechanisms underlying some of the key observations are not fully developed. For instance, the paper shows that intrinsic steering produces greater lexical diversity than prompted steering, but it is unclear whether this arises from properties of the value vectors themselves, or from differences in the underlying training signal.
5. The relationships between these Schwartz value directions and previously studied behavioral attributes are also underexplored. For example, attributes such as power seeking, risk seeking, wealth accumulation, or prosocial behavior have already been studied in prior interpretability and alignment work. It would strengthen the paper to clarify whether the proposed value vectors correspond to or subsume any of these previously identified behavioral directions.
6. Finally, the exact novelty is somewhat ambiguous. Extracting linear concept directions using difference in means estimators and intervening via activation steering is already well established in the literature. I endorse applying these techniques to Schwartz values is in my opinion novel enough an application in it's own right, in theory. However in execution the paper does not clearly articulate what new conceptual insight about value representations in LLMs emerges from this analysis. If the goal is to introduce a psychologically grounded framework for studying values in LLMs, the paper should more clearly develop the implications of the Schwartz value structure rather than treating it primarily as a labeling scheme for steering directions.
7. The value neurons component could be better grounded in previous literature, see for example [Universal Neurons in GPT-2 Language models](https://arxiv.org/abs/2401.12181)

Despite the issues with narrative framing and cohesion, the study introduces a psychologically grounded perspective on value representations in LLMs that could serve as a useful point for future work in this area, and is a original direction. At least the empirical claims and metholodgy appear technically sound and reasonably well supported, as well as clear in details, and this mitigates the narrative clunkiness.

### Suggestions:

1. The terms `value vectors` and `value expression` carry a lot of cognitive overload with other conventional uses of the word `value` in the context of mathematics, ethics and computing. Referring to `Schwarz` vectors or other such more conrete terminology might be helpful.

2. A more clear definition of the Schwartz circumplex would have been helpful for readers from non psychology background, perhaps as an appendix.

---

> ### Author Rebuttal · Authors · 2026-03-31
>
> Thank you for your thoughtful feedback.
>
> ## **Narrative of the paper (W1-3, Q2, Q3)**
>
> Although using the Schwartz value taxonomy is one of our contributions, our primary emphasis and core research question is to go one step further and investigate a methodology to understand the commonalities and differences between intrinsic and prompted value expressions and elucidate their effects and implications. We recognize that the transitions between sections lacked sufficient clarity, and we will improve the overall flow by restructuring the paper around these steps: first, methodology identifying the shared and unique components of intrinsic and prompted value expressions; second, behavior effects of the shared components and the circular Schwartz structure in LLM representations; and finally the behavior effects of unique components.
>
>
> We will revise the introduction to present the Schwartz value taxonomy more explicitly, and streamline the abstract by centering it on our core contribution which is investigating the dual mechanisms of value expression.
> ## **Causality of value vectors (W4)**
>
> Regarding whether the diversity difference between intrinsic and prompted value vectors arises from properties of the vectors themselves, or from differences in the underlying training signal, we believe it originates from the training signal, which is then captured by the value vectors. The reason is that, in our experiments (https://anonymous.4open.science/r/rebuttal-0B16/diversity_table.png), models showed lower response diversity when they were instructed to prioritize a certain value via a system prompt, compared to without such a system prompt. The same tendency was observed in the value vector steering experiments in our paper: lower response diversity with the prompted value vector than with the intrinsic vector; since the experiments involved no system prompt, this difference was caused solely by the two vectors. These results support that the diversity difference between intrinsic and prompted value vectors originates from the training signal and it is then captured by the value vectors.
> ## **Relationship with existing behavioral directions (W5)**
>
> Following your suggestion, we compared our value vectors with previously studied behavioral directions: prosociality and power-seeking.
>
> For prosociality, we used the following work: “Persona Vectors” [1]. We used the pre-computed altruism and forgiveness vectors implemented by Johnathan Sun [2]. Since both Persona Vectors and our method used Qwen2.5-7B-Instruct, the vectors are comparable. The results were intuitive and consistent with our expectations: our benevolence value vector showed higher cosine similarity with the prosocial vectors (altruism and forgiveness) than our power vector did. Specifically, prompted_benevolence scored 0.42 and 0.39 against altruism and forgiveness, respectively, while prompted_power scored only 0.23 and 0.09 against the same two directions. The same held for intrinsic vectors: intrinsic_benevolence scored 0.31 and 0.37, compared with 0.03 and -0.21 for intrinsic_power. The prompted variants were more similar to the Persona Vectors than the intrinsic variants were, which is expected because Persona Vectors were derived with persona prompts provided to the model.
>
> For the power-seeking behavior, we evaluated on the MACHIAVELLI benchmark [3] using Qwen2.5-7B-Instruct. We hypothesized that steering the model toward the Schwartz value of power would increase power-seeking behavior, whereas steering it toward benevolence would weaken the behavior. This is largely what we observe. Relative to the model without steering, steering with the power vectors increased the power-seeking score by 11.3% for prompted_power and 6.6% for intrinsic_power. It also increased the violation score, which reflects behaviors such as deception and manipulation, by 12.3% and 9.0%, respectively. In contrast, steering with intrinsic_benevolence reduced both the power-seeking score (-3.2%) and the violation score (-9.8%). prompted_benevolence showed the same pattern of reducing both the power-seeking score (-3.1%) and the violation score (-7.5%). Overall, the results support the expected semantic pattern: power vectors increase power-seeking and harmful behavior, whereas benevolence vectors weaken these tendencies.
>
> ## **Suggestions**
>
> We appreciate the reviewer's suggestion to rename "value" vectors. We agree the term could be misinterpretable, and will clarify early in the paper that it refers to human values to reduce ambiguity. We will also include a detailed explanation of the Schwartz circumplex in the appendix for readers from non-psychology backgrounds.
>
> [1] Chen et al. "Persona vectors: Monitoring and controlling character traits in language models", 2025.
>
> [2] https://github.com/johnathansun/persona-vector-agents
>
> [3] Pan et al. "Do the rewards justify the means? measuring trade-offs between rewards and ethical behavior in the machiavelli benchmark", 2023.

---

> > ### Author Rebuttal · Reviewer_As7k · 2026-04-02
> >
> > I do on balance like the paper; and would be pleased to see it accepted despite the current shortcomings. As reflected in my score of 4, and the strengths I have mentioned. I thank the author for their clarifications to my questions as well.
> >
> > But raising the score to an even higher rating of 5,  would require more substantial changes than feasible in rebuttal period. I also find the focus on "intrinsic vs prompted (Schwarz) value expressions" the authors answer is their core contribution, is not sufficiently supported by the paper narrative and experiments yet.
> >
> > It would have been more relevant to introduce the Schwarz taxonomy and the deeper study of how it is represented and influences model internals, rather than assuming this is apriori valid and jumping to the intrinsic vs prompted expression of it immediately. There may be a rewritten and reorganized version of this paper that assuages all my concerns, and the reordering the authors suggest is a promising one, but it would require substantial changes to implement.
> >
> > I will maintain my (positive) score.

---

> > > ### Author Response · Authors · 2026-04-06
> > >
> > > We appreciate that the reviewer on balance likes the paper and would be pleased to see it accepted.
> > >
> > > We agree that the remaining issue—refining the narrative and structure—is not easy to resolve in the short rebuttal. For the final paper, we are committed to introducing the Schwartz values upfront and establishing their internal representation as a foundational premise before moving onto the dual value mechanisms. This foundation will be grounded in (1) the diverse evidence we verified in the submitted manuscript, such as the circular geometric structure, cross-lingual consistency in value steering effectiveness, and the activations of value neurons in value-relevant contexts; and (2) the new experiments conducted for the rebuttal, which demonstrate the alignment of value vectors with prosocial and power-seeking behaviors on two external benchmarks, MACHIAVELLI and Persona Vectors.
> > >
> > > We thank the reviewer again for their time and valuable insights.

---

### Decision · Program_Chairs · 2026-04-30

**Decision:**

Accept (regular)

**Comment:**

This paper investigates how values are represented and expressed in LLMs using Schwartz’s psychological value taxonomy. It extracts value-specific activation directions and associated neurons, and studies how steering, ablation, and amplification along these directions affect model behavior, revealing both shared and distinct mechanisms between intrinsic and prompted value expression.

Reviewers raised several concerns, most of which have been largely addressed during the rebuttal. In particular, issues related to narrative clarity, robustness of methodological choices, and positioning relative to prior work were discussed and improved. After rebuttal, while some concerns remain regarding the smoothness and cohesion of the overall argument, all reviewers recommend acceptance. Therefore, I concur with this assessment and encourage the authors to revise the manuscript according to the discussion in the rebuttal for the final version.